# Pre-Hospital Pain Management in Children with Injuries: A Retrospective Cohort Study

**DOI:** 10.3390/jcm10143056

**Published:** 2021-07-09

**Authors:** Ada Holak, Michał Czapla, Marzena Zielińska

**Affiliations:** 1Department of Emergency Medicine, College of Rehabilitation, 01-234 Warsaw, Poland; ada.holak@wsr.edu.pl; 2Faculty of Health Sciences, Wroclaw Medical University, 51-618 Wrocław, Poland; 3Centre for Heart Diseases, University Hospital, 50-556 Wrocław, Poland; 4Department and Clinic of Anaesthesiology and Intensive Therapy, Faculty of Medicine, Wroclaw Medical University, 50-556 Wrocław, Poland; marzena.zielinska@gmail.com; 5Department of Paediatric Anaesthesiology and Intensive Care, University Hospital, 50-556 Wrocław, Poland

**Keywords:** acute pain, paediatric pain, injury, pain rating scale, emergency medical services, pain score, pre-hospital opioid analgesia

## Abstract

Background: The all-too-frequent failure to rate pain intensity, resulting in the lack of or inadequacy of pain management, has long ceased to be an exclusive problem of the young patient, becoming a major public health concern. This study aimed to evaluate the methods used for reducing post-traumatic pain in children and the frequency of use of such methods. Additionally, the methods of pain assessment and the frequency of their application in this age group were analysed. Methods: A retrospective analysis of 2452 medical records of emergency medical teams dispatched to injured children aged 0–18 years in the area around Warsaw (Poland). Results: Of all injured children, 1% (20 out of 2432) had their pain intensity rated, and the only tool used for this assessment was the numeric rating scale (NRS). Children with burns most frequently received a single analgesic drug or cooling (56.2%), whereas the least frequently used method was multimodal treatment combining pharmacotherapy and cooling (13.5%). Toddlers constituted the largest percentage of patients who were provided with cooling (12%). Immobilisation was most commonly used in adolescents (29%) and school-age children (*n* = 186; 24%). Conclusions: Low frequency of pain assessment emphasises the need to provide better training in the use of various pain rating scales and protocols. What is more, non-pharmacological methods (cooling and immobilisation) used for reducing pain in injured children still remain underutilized.

## 1. Introduction

Emergency teams (ETs) most frequently provide medical assistance to children with injuries at risk of pain related to the mechanism of such injuries. One of the most important interventions performed at the place of accident is to rate the intensity of pain and the adequacy of pain treatment. Failure to rate pain intensity and adequately manage acute pain leads to both immediate and long-term consequences. Among the short-term consequences should be mentioned, for example, delayed wound healing, while long-term consequences range from hypersensitivity to pain stimuli to the development of post-traumatic stress disorder (PTSD) [1,2]. The statistics clearly show that the large number of patients with undiagnosed and untreated pain is not just an individual issue but a common health problem of the general public [3].

Although there are barriers significantly limiting the measures used by ETs to reduce pain (lack of skills to obtain venous access, difficult pain assessment in preverbal children and toddlers, low experience of ETs in treating children, medical interventions at night, short transport time to hospital), the barriers should not justify untreated acute pain in children [4,5,6].

The widespread availability of age-appropriate pain rating scales, non-intravenous administration of effective drugs and non-pharmacological methods of pain relief, such as cooling and immobilisation, enable effective pain management as early as at the pre-hospital stage [7,8,9,10]. An injured child is usually attended to by an ET that is closest to the accident site.

This study aimed to evaluate the frequency, quality and methods of acute post-traumatic pain management in children at the pre-hospital stage.

## 2. Materials and Methods

### 2.1. Study Design and Setting

A retrospective study and analysis of medical records was conducted of patients with injury, aged between 1 month and 18 years, who were attended to by ETs. The primary intent of the analysis was to identify whether ETs assessed pain in children and whether traumatic pain was treated by them; if ETs treated traumatic pain, the study identified the drugs used in pain therapy and whether treatment included non-pharmacological agents in pain therapy, such as immobilization and cooling. Ambulances were equipped with analgesic drugs intended for administration by intravenous (intranasal mucosal atomization device was inaccessible), oral (tablets only) or rectal (suppositories and enemas) routes. The available immobilization equipment included: Kramer splints, orthopaedic board, c-spine collars and head blocks, Kendrick extrication device (KED), elastic and triangular bandages, and for cooling: burns dressings. The retrospective analysis covered medical records of 2452 injured patients, aged between 1 month to 18 years inclusive, who were attended to by ETs. The analysed ET interventions took place between 1 January 2016 and 31 December 2018 in an operational area around Warsaw (the capital city of Poland) with one million inhabitants. A total of 18 ETs, of which 5 were specialized ETs (S-ETs) comprising at least three persons, including a doctor and a medical nurse or paramedic, and 13 are basic ETs (B-ETs) comprising at least two persons qualified to perform medical rescue services, including a nurse or paramedic, were involved in those interventions [11].

### 2.2. Study Population

We analysed all the patients who met the inclusion criteria (diagnosis of injury, age 1 month to 18 years old). The medical records of a final group of 2452 patients were analysed. Medical records were selected containing diagnoses from the following groups according to the International Classification of Diseases 10 (ICD 10): (a) S00–S99 (diagnoses including an injury of the head, neck, chest, abdomen, lower back, lumbar spine and pelvis, shoulder and arm, elbow and forearm, wrist and hand, hip and thigh, knee and shin), (b) T00–T32 (injuries involving multiple body regions, unspecified parts of the trunk, extremities and body region, effects of foreign body penetration through natural body orifices, thermal and chemical burns, frostbite), (c) T90–T98 (sequelae of injuries, poisoning and other consequences of external causes, including animal bites).

### 2.3. Data Collection

The analysis included all data ambulance service medical documentation such as patients’ age, sex, mechanism of injury, accident site, vital signs (respiratory rate, blood pressure, heart rate, saturation), decision concerning transport to hospital and information concerning the treatment used (use of pain rating scale, pharmacotherapy including the route of administration and possible complications, immobilisation, cooling). The low-energy injury mechanism included a hit or a fall from the height of one’s own body, while the high-energy included traffic accident, a fall from a height greater than the victim, a blow from an object, beating or biting by an animal. During the study period in Poland, there were no formal protocols in force for rating pain intensity and for pain management. ETs had no pain rating scale imposed. They could freely choose from among the scales known to them from the numerical and behavioural groups.

### 2.4. Ethical Considerations

This study was approved by the independent Bioethics Committee of the Wroclaw Medical University (decision no. 424/2018). The study was carried out in accordance with the tenets of the Declaration of Helsinki and recommendations of good clinical practice. For reporting, the Strengthening the Reporting of Observational Studies in Epidemiology (STROBE) guidelines were followed.

### 2.5. Statistical Analysis

The statistical analysis was performed using Statistica 13 software (TIBCO, Inc., Palo Alto, CA, USA). Arithmetic means, medians, standard deviations and the range of variability (extreme values) were calculated for measurable variables. In the case of qualitative variables, their frequency (%) was calculated. All analysed quantitative variables were verified, with the Shapiro–Wilk test used to determine the type of distribution. Qualitative variables were compared between groups using the chi-squared test (χ2). The results were considered statistically significant at *p* < 0.05.

## 3. Results

### 3.1. Characteristics of the Study Group

The study group (*n* = 2452) consisted of 898 females, representing 37% of the entire group, and 1554 males (63%). The age across the group ranged from 1 month to 18 years (mean ($\bar{x}$) = 10.0 years; SD = 5.3 years).

Five age categories were identified: infants (aged 0–12 months), toddlers (aged 1–3), preschool-age children (aged 4–6), school-age children (aged 7–12) and adolescents (aged 13–18). In the analysed group, adolescents represented more than 40% of injured patients (*n* = 1022), followed by school-age children (*n* = 787; 32%). Preschool-age children and toddlers represented approximately 11% of injured patients. Infants were the least numerous age group (*n* = 92, 4%). In the analysed group, ETs provided medical assistance most frequently to patients with low-energy injuries: 65% (*n* = 1606). A total of 31% of cases (*n* = 737) were high-energy injuries and 4% of ET interventions involved child burn victims (*n* = 89).

Dispatch location for ET, medical history and description of circumstances of the accident, which were included in medical records, indicated that the most common accident site was home (*n* = 838, 34%), followed by school (*n* = 699, 29%), public places (*n* = 492, 20%) and street and road traffic (*n* = 410, 17%). A comparative analysis by type of injury was conducted. The prevalence of a specific type of injury depended on factors such as sex, age group and accident site.

Low-energy injury was the most common mechanism of injury in both males and females (males: *n* = 1048; females: *n* = 556). All types of injuries were more common in males compared with females. Adolescents were the group most likely to be injured, with as much as 40% of low- and high-energy injuries occurring in that age group. More than 60% of injuries in adolescents were burn-related. Infants and toddlers were the least likely to sustain injuries.

Burn-related injuries most frequently occurred at home, low-energy injuries at school, while high-energy injuries were most common in street and road traffic (Table 1).

### 3.2. The Use of Pain Rating Scales According to the Type of ET and Undertaken Medical Intervention

ETs were free to choose a pain rating scale. In 99% of cases, no pain rating scale was used (*n* = 2432). Pain was rated in 20 patients (1% of the study population), with NRS used as the assessment tool in all of them. Behaviour rating scales (BRS) were not administered to any patient. The type of ET providing medical assistance was nonsignificant (Table 2).

### 3.3. Pharmacological Treatment and Transport to Hospital by Type of Intervening ET

Of all tested children, 16% received some kind of drug. From the group of analgesics including acetaminophen, ibuprofen, acetylsalicylic acid, ketoprofen, metamizole, fentanyl and morphine, the drugs that were most commonly administered to children were acetaminophen (*n* = 107, 4%) and morphine (*n* = 161, 7%). None of ETs administered acetylsalicylic acid to children. As far as sedatives are concerned, diazepam and midazolam were used.

Acetaminophen was most commonly used in children with burns (*n* = 16, 29%)—more frequently than in low- and/or high-energy injuries. Morphine was administered to 40% of children with burns, 5% of children with high-energy injuries and 5% of children with low-energy injuries. Those differences were statistically significant, just as in the case of ibuprofen intake (Table 3).

S-ETs were statistically significantly more likely to administer drugs such as acetaminophen, metamizole, midazolam and diazepam. (Table 4).

The vast majority of injured patients (70%) were transported to the hospital (*n* = 1723). Over 70% of interventions were carried out by B-ETs (*n* = 1820, 74%). This result may have been affected mainly by both the disproportion between the number of B- and S-ETs in the analysed area (72%—B-ETs, 28%—S-ETs) and the dispatching system which shows the nearest available ET to the emergency medical dispatcher.

### 3.4. Non-Pharmacological Methods of Analgesia Used by ETs

Non-pharmacological techniques, such as immobilisation and cooling, help reduce pain and are recommended as adjunctive therapy with pharmacological treatment. Immobilisation was applied in 22% (*n* = 538), while cooling in 3% of cases (*n* = 62). S-ETs were more likely to use cooling (5% of all interventions) than B-ETs (2% of all interventions). Immobilisation, on the other hand, was statistically more frequently used by B-ETs (24% of all B-ET interventions) compared with S-ETs (16% of all S-ET interventions) (Table 5).

Cooling was statistically significantly more common in the case of burns (*n* = 56, 63%). In the case of low- and high-energy injuries, the number of patients who received cooling was very small—4 and 2 patients, respectively. Those differences were statistically significant. When it comes to immobilisations, the highest number of this type of interventions was used for low- (22%) and high-energy injuries (23%). In the case of burns, immobilisation was used in only 3 patients (*n* = 3) (Table 6).

B-ETs were by far most frequently dispatched to patients with low-energy injuries (71% of all ET interventions). S-ET interventions usually concerned low- (49%) and high-energy injuries (43%). Accident site also affected the type of ET dispatched by the Emergency Communication Centre (ECC). In the case of S-ETs, most interventions occurred at home (37%), while B-ETs provided medical assistance at home (33%) with approximately the same frequency as at school (33%).

The use of comprehensive analgesic treatment by ETs combining pharmacotherapy and cooling was also analysed. The use of cooling was statistically significant mainly in burns, as was the use of pharmacologic analgesia involving one or more drugs. A very high percentage of patients with low- and high-energy injuries (84.4% and 83.5%, respectively) did not receive any analgesic drug. One-third of children with burns did not receive analgesia, and more than half of them were provided with an analgesic drug or wound cooling. Comprehensive treatment was used only in 13.5% of cases (Table 6).

### 3.5. The Assessment of Differences in the Treatment of the Same Injuries across Age Groups

There were differences in relationships between age category and sex or accident site. Male patients predominated in each age category. High-energy injuries were most common in infants, while low-energy injuries predominated in other age groups. As regards the accident site, infants, toddlers and preschool-age children were most frequently injured at home (>50%). In other age categories, school was the most common accident site (Appendix A).

The frequency of use of non-pharmacological methods of pain management in each age group was also rated. Toddlers represented the largest percentage of patients who were provided with cooling (*n* = 34, 12%), while immobilisation was most commonly used in adolescents (*n* = 294, 29%) and school-age children (*n* = 186; 24%) (Appendix A).

The analysis of the frequency of the use of individual analgesics in selected age groups of patients revealed statistically significant differences in the use of a particular type of analgesic depending on patient’s age. Acetaminophen, ketoprofen, metamizole and fentanyl were drugs statistically significantly more often administered to older children (school-age children and adolescents) (Appendix A). The sedative drug diazepam was statistically more commonly used in school-age children and adolescents.

## 4. Discussion

The problem of inadequate analgesic treatment of children is of interest to an increasing number of scientists, but in the available literature it is most often about finding barriers to diagnosis and treatment without documenting the numerical scale of the problem.

An analysis of the presented material showed that variables such as the patient’s age, sex, accident site and the decision about transport to the hospital did not affect the frequency of pain assessment during the study period. When comparing the results of the presented study with other available results, the difference in the number of performed pain assessments is surprising. It should be emphasized that ETs did not have an imposed pain evaluation protocol. They could freely choose from among the scales known to them from the numerical and behavioural groups.

According to the present study, pain intensity was rated only in 1% of all interventions provided by ETs to children with acute post-traumatic pain. Brown et al. presented significantly better results: in their study, pain intensity was rated in 25% of children. In a study by Murphy et al., however, as many as 32% of children had their pain intensity rated [12,13,14]. Rahman et al. investigated a group of 202 EMS workers for the city of Edmonton, of whom 94% completed the survey. In self-assessment questionnaires, as many as 62% of them stated that they always used a pain rating scale of their choice in children: NRS for adolescents (96%) and BRS for children (57%). It turned out, however, that those statements were only declarations since a retrospective analysis prepared by the same team of researchers showed that in a group of 696 paediatric patients treated by the same EMS workers, pain was not documented in as many as 86.6% of them [15].

It should be emphasized that during the analysed period, Polish ETs were not equipped with any protocols for pain assessment and pain management. When comparing the period before and after the implementation of such a protocol, Brown et al. found that its implementation did not increase the percentage of children with documented pain assessment or the frequency of analgesic intake [12,13]. Similar findings are presented by Murphy et al. In their study, pain assessment was the same before and after the implementation of a similar protocol and amounted to 18% [14]. Jaeger et al. analysed the existence of a correlation between the implementation of a pain assessment protocol in medical records and the frequency of drug administration. Over a period of 13 months, they observed an increase in the number of cases with documented pain (from 25% to 100%). However, they also found no impact of the protocols used on the increase in the frequency of the use of analgesics [16]. Despite the existence of a protocol, pain is not always assessed, e.g., in the Lord et al. study 18.8% of patients have not had such an assessment [17].

In the group of children with acute post-traumatic pain and rated pain intensity, 55% of patients received analgesics, while in the group of children without pain assessment, only 16% received analgesia. The assessment of pain intensity significantly increased the intake of ibuprofen (*p* = 0.004) and opioids such as morphine (*p* < 0.001) and fentanyl (*p* < 0.001). Despite many scientific reports on the efficacy of nasal administration of fentanyl, none of the analysed ETs administered it by this route [18,19,20].

In a study by Hewes et al., 19.9% of all patients with documented post-traumatic pain received analgesics [21]. Even patients with severe pain (VNRS 8–10) do not always receive any analgesia—the Lord et al. study showed 55% were untreated children and the Pilbery et al. study showed 87% [17,22].

A 2-fold increase in opioid intake in a group of children with rated pain was observed by Brown et al. At the same time, Brown et al. stressed that the implementation of a pain assessment protocol did not increase the frequency of pain assessment, but it significantly increased both the total administered dose of morphine and the dose calculated per kg body weight (by 18% and 14.9%, respectively). In other words, the drugs started to be administered at appropriate doses [12].

An analysis by Murphy et al. shows that 32% of children with pain documented by the Emergency Medical Service (EMS) were transported to hospital. A total of 26% of them received an analgesic drug. In this group, acetaminophen was administered in 35%, ibuprofen in 23% and inhaled methoxyflurane in 11% of cases [14].

The drugs most commonly administered by ETs to injured children included morphine (7%) and acetaminophen (4%) (*p* < 0.001). Children with burns received as many drugs as possible (morphine—40%, acetaminophen—29%). The above-mentioned mechanism of injury also significantly increased the frequency of administration of sedatives such as diazepam (8%, *p* < 0.001) and midazolam (3%, *p* < 0.001). A study by Nadolny et al. showed that those drugs were administered in 16.5% of all interventions involving children with burns [23]. Similar conclusions were presented in a study by Rahman et al., who showed that analgesics were most frequently administered to children with burns and less frequently to children with muscle, bone and joint injuries. The risk of burns is higher especially in toddlers whose spatial awareness and psychomotor skills are not fully developed [24,25].

Both types of ETs were equipped with the same set of drugs used for analgosedation. During interventions, S-ETs used such drugs more frequently compared with B-ETs. This is due to the mechanism of injury to which ETs were dispatched. S-ETs most frequently provided medical assistance to patients with burns, while B-ETs were dispatched to children with high- and low-energy injuries. Non-opioid analgesics (acetaminophen) and non-steroidal anti-inflammatory drugs (NSAIDs—metamizole) were drugs preferred by S-ETs. According to the collected study material, opioid analgesics such as morphine and fentanyl were administered equally frequently by both types of ETs, which is why their intake was more affected by the mechanism of injury and the use of a pain rating scale than by the type of ET. S-ETs tend to use multimodal analgesia, especially analgosedation. Midazolam was administered by S-ETs 10 times more frequently than by B-ETs, whereas the administration of diazepam by S-ETs was 8 times more frequent than by B-ETs. As Bayat rightly points out it should be noted that although benzodiazepines have an excellent sedative effect, they cannot be used as a substitute for analgesics [26].

In an analysis of the non-pharmacological methods of pain relief, a statistical survey of the analysed medical records of the entire study population showed that immobilisation was used in 22% of cases, while cooling was used in only 3% of all interventions; however, in the case of patients with burns, cooling was used in as many as 63% of cases. Whitley shows that 9.6% of children with injuries from his study population received other forms of non-pharmacological pain relief such as splinting, dressing or slings [27]. Strobel and Baartmans, when describing the rescue management in children with burns, note the importance of cooling in burn wounds. Even covering the burnt area of skin with wet gauze reduces pain. Nevertheless, excessive use of cooling carries the risk of hypothermia, which is why, when treating children with burn wounds, the principle “cool the burn and keep the patient warm” should be followed [28,29]. ETs applied immobilisation in the case of low- and high-energy injuries (22% and 23%, respectively). Cooling was never used in those mechanisms of injury. A much lower percentage of immobilised and cooled low- and high-energy injuries is presented by Izsak et al. In the study population, immobilisation was used in 7.2%, while cooling was applied in 1.7% of cases [30].

As noted by Häske et al., immobilisation of fractured limbs is of analgesic value, especially in displaced fractures or joint dislocations in which perfusion abnormalities occur and the risk of necrosis of the surrounding tissues increases. However, pharmacological anaesthesia should be taken into consideration before the affected limb is repositioned in its longitudinal axis using traction. This is because, in many cases, the use of immobilisation itself will be insufficient to reduce pain. Topical cryotherapy may also be used even though there is low evidence for its effectiveness. Non-pharmacologic methods of pain relief should not be prioritised in the case of life-threatening injuries [31]. At the same time, post-traumatic immobilisation should not be completely neglected even though—due to differences in the structure of the skeletal system of a child—such measures affect the further process of injury healing [32].

### Limitations

Our study has several limitations, principally related to the EMS documentation available to us for analysis. The main shortcoming was the lack of reported time at which the patient’s pain intensity was rated by ETs. Additionally, there was no information on whether the use of non-pharmacological methods of pain management was preceded by the assessment or, on the contrary, whether the pain was rated only after such methods were applied.

It was also impossible to determine the criterion according to which a B- or S-ET was dispatched to the injured patient. According to generally applicable rules, an emergency medical dispatcher who sees the location of ambulances on maps should dispatch a team that can arrive at the accident site within the shortest possible time. The disproportion between the number of specific types of ETs in the analysed operational area (72%—B-ETs, 28%—S-ETs) should also be taken into consideration in this analysis as it may directly translate into the medical intervention undertaken, especially for patients with serious life-threatening injuries. Future studies need to be designed to address these important outcomes and to improve the documentation of pain management within the EMS system.

## 5. Conclusions

Neither basic nor specialist emergency medical teams rate pain intensity in paediatric patients using pain rating scales. In individual cases, the numeric rating scale is applied. Behaviour rating scales are not used by ETs. In post-traumatic pain management in children, opioids are used equally frequently by both types of ETs, while multimodal therapy is used more frequently by S-ETs.

The decision about the administration of analgesics in children with post-traumatic pain is mainly affected by the mechanism of injury (burns). It is not, however, affected by factors such as the patient’s age (except for infants), the decision on whether or not to transport the patient to the hospital or abnormal vital parameters.

Toddlers with burns, in whom analgosedation combined with cooling is frequently used, are the patient group that is the best protected from pain. Cooling is applied by ETs only for burns/scalds, and it is not used for treating other injuries. The analysis confirmed the low incidence of pain assessment by the ETs, which further supports the need to continue training among medical staff.

## Figures and Tables

**Table 1 jcm-10-03056-t001:** Characteristics of the study group and a comparison of the number of patients with a specific injury according to sex, age and accident site.

Study Group *n* = 2452	Mean			Me		Min–Max	SD
Age (Years)	10.0			10.9		0.1–10.0	5.3
Sex	Female			37%		*n* = 898	
	Male			63%		*n* = 1554	
Variables		Injury					
		Low-Energy		High-Energy		Burn-Related	
		*n*	%	*n*	%	*n*	%
Sex	Male	1048	65	439	58	66	74
	Female	556	35	318	42	23	26
Age	Infants	37	2	51	7	57	64
	Toddlers	144	9	72	10	14	16
	Preschool-age children	181	11	83	11	6	7
	School-age children	560	35	221	29	8	9
	Adolescents	684	43	330	44	57	64
Accident site	Home	565	35	196	26	77	87
	School	606	38	90	12	3	3
	Street and road traffic	53	3	353	47	4	4
	Agriculture	3	0	4	1	-	-
	Public place	374	23	113	15	5	6
	Work	2	0	-	-	-	-

*n*—number of patients; %—percentage of patients; Me—median; min—minimum value; max—maximum value; SD—standard deviation.

**Table 2 jcm-10-03056-t002:** The frequency of use of pain score documented by different types of ETs.

		ET				*p*-Value *
		Specialist		Basic		
		*n*	%	*n*	%	
Pain score documented	No	628	99	1804	99	*p* = 0.55
	Yes	4	1	16	1	

* χ^2^ test; n—number of patients; %—percentage of patients; ET—emergency team.

**Table 3 jcm-10-03056-t003:** The intake of individual analgesics and sedatives by mechanism of injury.

		Injury						*p*-Value *
		Low-Energy		High-Energy		Burn-Related		
		*n*	%	*n*	%	*n*	%	
Acetaminophen	No	1549	96	733	97	63	71	*p* < 0.001
	Yes	57	4	24	3	26	29	
Ibuprofen	No	1601	100	754	100	85	96	*p* < 0.001
	Yes	5	0	3	0	4	4	
Acetylsalicylic acid	No	1606	100	757	100	757	100	*p* = 1.00
	Yes	-	-	-	-	-	-	
Ketoprofen	No	1567	98	729	96	88	99	*p* = 0.13
	Yes	39	2	28	4	1	1	
Metamizole	No	1588	99	750	99	89	100	*p* = 0.56
	Yes	18	1	7	1	-	-	
Fentanyl	No	1575	98	742	98	89	100	*p* = 0.41
	Yes	31	2	15	2	-	-	
Morphine	No	1519	95	719	95	53	60	*p* < 0.001
	Yes	87	5	38	5	36	40	
Midazolam	No	1605	100	749	99	86	97	*p* < 0.001
	Yes	1	0	8	1	3	3	
Diazepam	No	1603	100	753	99	82	92	*p* < 0.001
	Yes	3	0	4	1	7	8	

* χ^2^ test; *n*—number of patients; %—percentage of patients.

**Table 4 jcm-10-03056-t004:** The intake of individual analgesics and sedatives administered by specialist and basic ETs.

		ET				*p*-Value *
		Specialist		Basic		
		*n*	%	*n*	%	
Acetaminophen	No	584	92	1761	97	*p* < 0.001
	Yes	48	8	59	3	
Ibuprofen	No	628	99	1812	100	*p* = 0.54
	Yes	4	1	8	0	
Acetylsalicylic acid	No	632	100	1820	100	*p* = 1.00
	Yes	-	-	-	-	
Ketoprofen	No	610	97	1774	97	*p* = 0.20
	Yes	22	3	46	3	
Metamizole	No	618	98	1809	99	*p* = 0.001
	Yes	14	2	11	1	
Fentanyl	No	616	97	1790	98	*p* = 0.16
	Yes	16	3	30	2	
Morphine	No	581	92	1710	94	*p* = 0.08
	Yes	51	8	110	6	
Midazolam	No	622	98	1818	100	*p* < 0.001
	Yes	10	2	2	0	
Diazepam	No	624	99	1814	100	*p* = 0.007
	Yes	8	1	6	0	

* χ^2^ test; *n*—number of patients; %—percentage of patients; ET—emergency team.

**Table 5 jcm-10-03056-t005:** The frequency of immobilisation and cooling used by different types of ETs.

		ET				*p*-Value *
		Specialist		Basic		
		*n*	%	*n*	%	
Cooling	No	601	95	1789	98	*p* < 0.001
	Yes	31	5	31	2	
Immobilisation	No	529	84	1385	76	*p* < 0.001
	Yes	103	16	435	24	

* χ^2^ test; *n*—number of patients; %—percentage of patients; ET—emergency team.

**Table 6 jcm-10-03056-t006:** An analysis of the relationship between the mechanism of injury and the use of non-pharmacological methods or drugs of analgesia.

		Low-Energy Injury (*n* = 1606)		High-Energy Injury (*n* = 757)		Burn-Related (*n* = 89)		*p*-Value *
		*n*	%	*n*	%	*n*	%	
Cooling	No	1602	99.8	755	99.7	33	37.1	*p* < 0.001
	Yes	4	0.2	2	0.3	56	62.9	
Immobilisation	No	1246	7.6	582	76.9	86	96.6	*p* < 0.001
	Yes	360	22.4	175	22.1	3	3.4	
Number of drugs	0	1356	84.4	632	83.5	24	26.9	*p* < 0.001
	1	204	12.7	97	12.8	47	52.8	
	More than 1	46	2.9	28	3.7	18	20.3	
Cooling and/or a drug	None	1558	97.0	727	96.0	27	30.3	*p* < 0.001
	Cooling or a drug	46	2.9	30	4.0	50	56.2	
	Cooling and a drug	2	0.1	-	-	12	13.5	

* χ^2^ test; *n*—number of patients; %—percentage of patients.

## Data Availability

The data will be available by contacting the corresponding author.

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
