# Peer review of "Pre-Hospital Pain Management in Children with Injuries: A Retrospective Cohort Study"

_jcm, 2021, doi:10.3390/jcm10143056_

Round 1

Reviewer 1 Report

I found the writing to be much improved over the last version.  One error still stood out - at the end of the Abstract, the word was changed from "underappreciated" to "underestimated", which I still think is inappropriate.  If the intention is to point out that these non-pharmacological methods are not being used (which is the way I interpret it), an appropriate word would be "underutilized".  If, on the other hand, the intention is to say that the methods need to be studied more, an appropriate set of words would be "need further study" in place of "still remain underestimated".

Under the description of high-energy mechanism, it lists "a blow to an object".  I am not sure what this means.  Is it a blow from an object?

The authors stated that the minimum age was fixed in Table 1.  However, it appears to be the same.  Either the minimum age in months is incorrect or the minimum age in years is incorrect, because 1.0 months can never be equal to 0.0 years, despite any value before rounding (0.0 years would be at most 0.05 years before rounding, which equals at most 0.7 months).

There are still too many tables.

Author Response

Thank you for your kind feedback

  1. I found the writing to be much improved over the last version.  One error still stood out - at the end of the Abstract, the word was changed from "underappreciated" to "underestimated", which I still think is inappropriate.  If the intention is to point out that these non-pharmacological methods are not being used (which is the way I interpret it), an appropriate word would be "underutilized".  If, on the other hand, the intention is to say that the methods need to be studied more, an appropriate set of words would be "need further study" in place of "still remain underestimated".
    Reply: Thank you very much for this remark. The intention is to point out that these non-pharmacological methods are not being used. We corrected for word "underutilized".
  2. Under the description of high-energy mechanism, it lists "a blow to an object". I am not sure what this means. Is it a blow from an object?
    Reply: Thank you very much for this remark. You are right, we meant “a blow from an object”. We corrected it.
  1. The authors stated that the minimum age was fixed in Table 1. However, it appears to be the same.  Either the minimum age in months is incorrect or the minimum age in years is incorrect, because 1.0 months can never be equal to 0.0 years, despite any value before rounding (0.0 years would be at most 0.05 years before rounding, which equals at most 0.7 months).
    Replay: Thank you very much for this comment. You’re right. In the new version of manuscript it has been corrected.

     4. There are still too many tables.

Reply: Thank you very much for this remark. We reduced the number of tables to 6 in the main text and 3 in a supplementary file.

Reviewer 2 Report

Thank you for giving me the opportunity to read this manuscript, titled “Pre-hospital pain management in children with injuries: A retrospective cohort study.”  This is an important study that aims to evaluate methods of traumatic pain assessment and management in children in the pre-hospital EMS setting in Poland.  I think overall it is well written and has some very interesting findings.  I have one main comment, followed by a number of minor suggestions.

Main Comment:

My main comment is regarding the number of tables. I think the use of 12 tables is too much, it dilutes your key findings/message.  Your main aim is to evaluate methods for traumatic pain management, and to evaluate pain assessment.  I feel that 12 tables, whilst useful, is too much.  It might be beneficial to the reader to focus on fewer tables, perhaps 6 or less.  You could look to combine/condense tables together, or add them as a supplementary file perhaps if the journal allows?

Minor suggestions:

Abstract:

Line 25-26: It is unclear why p values are needed in the abstract when you are not describing comparators.

Introduction:

Line 37: EMT is internationally recognised as “emergency medical technician”.  It might be confusing for some readers to refer to EMT throughout when you mean “emergency medical team”.  If there is an alternative, I would suggest using it.  If not, then maybe consider abbreviating just the “emergency medical” part, and refer to “EM team” throughout.

Line 47: There is a recent systematic review for the barriers to pain management in children in the pre-hospital setting you might find useful (https://doi.org/10.1177%2F1367493520949427)

Methods:

Line 60: It would be useful to explain in the setting, which analgesics are available to B-EMTs and S-EMTs, the presentation of the analgesics (tablet/syrup/powder/vial) and route available (oral, IV, IN etc.).  Also explain the non-pharmacological treatments “immobilisation” and “cooling”, what do you mean specifically? Slings, splints, c-spine collars and head blocks, ice packs, burns dressings?

Line 62: It would be useful to be clear about the age range.  Are 1 month and 18-year-old patients included?  You might consider “1 month to 18 years, inclusive”.

Line 110:  Just a suggestion, but you could use the term “categorical variables” rather than “qualitative variables”.  Also, you could use “numerical variables” instead of “quantitative variables”. 

Results:

Line 116-117: It would be better to refer to patients as male or female throughout, rather than girls or boys.

Table 1:  It would be better to use the word “mean” rather than the symbol, and the word “median” rather than “me” within the table.  You should also consider combining the mean and standard deviation and report them on the same row: “Mean (SD) | 10.9 (5.3)”.  When reporting the median you should also report the interquartile range. I’m not sure age in months is needed.

Table 2: Comparing characteristics to injury type doesn’t answer your research question.  Consider combining this with table 1 to give a more comprehensive patient characteristics table (sex, age, incident location, type of injury, pain score documented (yes/no), analgesic/sedative administered (yes/no)).

Table 3: This is important as it answers one of your main questions.  Instead of “pain rating scale” perhaps say “Pain score documented”.

Table 4: Maybe have a total row at the end, so we can see the total percentage of children who received a drug.

Table 5: Again, maybe a total row at the bottom of the table.  Also, should “Midanium” be “Midazolam”?

Table 6: This is an interesting table, again a total row at the bottom.  I wonder, is there a way to combine all these analgesic tables?  Maybe switch the axis (so to speak) and have drugs across the top, and all the different various groups across the side as rows?  Just a consideration. 

Line 182: I think you mean to say “Non-pharmacological techniques, such as immobilisation and cooling, help reduce pain and are recommended…”

Table 7, 8 and 9:  I think there is potential to combine these, perhaps have Immobilisation and cooling across the top, and then all your categories and p-values down the side as rows.

Table 10, 11 and 12 maybe these could be added as supplementary data if the journal allows?  I’m conscious that you have so many tables and it’s a little information overload for the reader.

Discussion.

A key concern is that pain isn’t being assessed/measured (1%), therefore you don’t know how effective the pain management strategies are.  You can’t determine rates of effective pain management, as you don’t have two pain scores (pre and post intervention) and therefore you are missing a key quality measure (rate of effective pain management, often defined as a pain score reduction of 2 or more out of 10).  I think it would be helpful to discuss this, along with possible solutions (electronic clinical records with mandatory pain scoring, for example, and addition of age appropriate pain scales, as discussed below).

Line 270: Whose data are you discussing? Yours? Can you be a little clearer in this paragraph, as you’ve been discussing other studies in the previous paragraph.

A common pain scale used for children (aged 3 to 7 approximately) is the Wong and Baker FACES scale.  Do Polish paramedics have access to such a scale?  This might help with the pain assessment problem.  A behavioural pain scale such as FLACC for pre-verbal children, a faces scale for children aged 3-7 and then the numeric scale for older children aged 8+.  Perhaps a suggestion for future improvement?

Line 358: Remove the number 5 next to conclusion.

Author Response

REVIEWER 2

Thank you for giving me the opportunity to read this manuscript, titled “Pre-hospital pain management in children with injuries: A retrospective cohort study.”  This is an important study that aims to evaluate methods of traumatic pain assessment and management in children in the pre-hospital EMS setting in Poland.  I think overall it is well written and has some very interesting findings.  I have one main comment, followed by a number of minor suggestions.

Replay: Thank you for your kind feedback.

  1. My main comment is regarding the number of tables. I think the use of 12 tables is too much, it dilutes your key findings/message.  Your main aim is to evaluate methods for traumatic pain management, and to evaluate pain assessment.  I feel that 12 tables, whilst useful, is too much.  It might be beneficial to the reader to focus on fewer tables, perhaps 6 or less.  You could look to combine/condense tables together, or add them as a supplementary file perhaps if the journal allows?

- Reply: Thank you very much for this remark. We reduced the number of tables to 6 in the main text and 3 in a supplementary file.

  1. Line 25-26: It is unclear why p values are needed in the abstract when you are not describing comparators.

- Reply: Thank you very much for this comment. You’re right. In the new version of manuscript it has been corrected.

  1. Introduction:

- Line 37: EMT is internationally recognised as “emergency medical technician”.  It might be confusing for some readers to refer to EMT throughout when you mean “emergency medical team”.  If there is an alternative, I would suggest using it.  If not, then maybe consider abbreviating just the “emergency medical” part, and refer to “EM team” throughout.

- Reply: Thank you very much for this comment. You’re right. In the new version of manuscript it has been corrected. We used basic emergency team (B-ET) and specialized emergency team (S-ET).

- Line 47: There is a recent systematic review for the barriers to pain management in children in the pre-hospital setting you might find useful (https://doi.org/10.1177%2F1367493520949427)

- Reply: Thank you very much for this comment. You’re right. In the new version of manuscript it has been added.

  1. Methods:

Line 60: It would be useful to explain in the setting, which analgesics are available to B-EMTs and S-EMTs, the presentation of the analgesics (tablet/syrup/powder/vial) and route available (oral, IV, IN etc.).  Also explain the non-pharmacological treatments “immobilisation” and “cooling”, what do you mean specifically? Slings, splints, c-spine collars and head blocks, ice packs, burns dressings?

- Reply: Thank you very much for this remark. In the new version of manuscript it has been corrected.

Line 62: It would be useful to be clear about the age range.  Are 1 month and 18-year-old patients included?  You might consider “1 month to 18 years, inclusive”.

- Reply: Thank you very much for this remark. In the new version of manuscript it has been corrected.

Line 110:  Just a suggestion, but you could use the term “categorical variables” rather than “qualitative variables”.  Also, you could use “numerical variables” instead of “quantitative variables”.

- Reply: Thank you very much for this suggestion. We prefer to leave it in the original version.

  1. Results:

Line 116-117: It would be better to refer to patients as male or female throughout, rather than girls or boys.

- Reply: Thank you very much for this comment. In the new version of manuscript it has been corrected.

Table 1:  It would be better to use the word “mean” rather than the symbol, and the word “median” rather than “me” within the table.  You should also consider combining the mean and standard deviation and report them on the same row: “Mean (SD) | 10.9 (5.3)”.  When reporting the median you should also report the interquartile range. I’m not sure age in months is needed.

- Reply: Thank you very much for this comment. The table has been corrected.

Table 2: Comparing characteristics to injury type doesn’t answer your research question.  Consider combining this with table 1 to give a more comprehensive patient characteristics table (sex, age, incident location, type of injury, pain score documented (yes/no), analgesic/sedative administered (yes/no)).

- Reply: Thank you very much for this comment. The tables has been merged.

Table 3: This is important as it answers one of your main questions.  Instead of “pain rating scale” perhaps say “Pain score documented”.

- Replay: Thank you very much for this comment. You’re right. In the new version of manuscript it has been corrected.

Table 4: Maybe have a total row at the end, so we can see the total percentage of children who received a drug.

- Replay: Thank you very much for this comment. You’re right. In the new version of manuscript it has been added in the chapter 3.3 – line 190.

Table 5: Again, maybe a total row at the bottom of the table.  Also, should “Midanium” be “Midazolam”?

- Reply: Thank you very much for this remark. In the new version of manuscript it has been corrected.

Table 6: This is an interesting table, again a total row at the bottom. I wonder, is there a way to combine all these analgesic tables?  Maybe switch the axis (so to speak) and have drugs across the top, and all the different various groups across the side as rows?  Just a consideration.

- Replay:  Thank you very much for this comment. Due to the large number of drugs exchanged, the table with them at the top, in the columns becomes less clear.

Line 182: I think you mean to say “Non-pharmacological techniques, such as immobilisation and cooling, help reduce pain and are recommended…”

- Replay: Thank you very much for this comment. You’re right. In the new version of manuscript it has been corrected.

Table 7, 8 and 9:  I think there is potential to combine these, perhaps have Immobilisation and cooling across the top, and then all your categories and p-values down the side as rows.

  • Replay: Thank you very much for this comment. We transformed tables 7 and 8.

Table 10, 11 and 12 maybe these could be added as supplementary data if the journal allows?  I’m conscious that you have so many tables and it’s a little information overload for the reader.

- Reply: Thank you very much for this remark. In the new version of manuscript it has been moved to the appendix.

  1.  A key concern is that pain isn’t being assessed/measured (1%), therefore you don’t know how effective the pain management strategies are.  You can’t determine rates of effective pain management, as you don’t have two pain scores (pre and post intervention) and therefore you are missing a key quality measure (rate of effective pain management, often defined as a pain score reduction of 2 or more out of 10).  I think it would be helpful to discuss this, along with possible solutions (electronic clinical records with mandatory pain scoring, for example, and addition of age appropriate pain scales, as discussed below).
  2. A common pain scale used for children (aged 3 to 7 approximately) is the Wong and Baker FACES scale.  Do Polish paramedics have access to such a scale?  This might help with the pain assessment problem.  A behavioural pain scale such as FLACC for pre-verbal children, a faces scale for children aged 3-7 and then the numeric scale for older children aged 8+.  Perhaps a suggestion for future improvement?

- Reply: Thank you very much for this remark. we clarified and emphasized that ET did not have an imposed pain evaluation protocol. They could freely choose from among the scales known to them from the numerical and behavioural groups.

Line 270: Whose data are you discussing? Yours? Can you be a little clearer in this paragraph, as you’ve been discussing other studies in the previous paragraph.

- Replay: Thank you very much for this comment. You’re right. In the new version of manuscript it has been corrected.

This manuscript is a resubmission of an earlier submission. The following is a list of the peer review reports and author responses from that submission.

Round 1

Reviewer 1 Report

The authors evaluated the methods used for reducing post-traumatic pain in children and their frequency of use, as well as the methods of rating pain and the frequency of their application in this age group.

The methodology as well as study design are not good. Basic methodological informations are not presented.

My concerns are as follows:

  1. Whether the study was approved by Ethic Committee? This should be clearly stated in methodology, together with institutional review board reference
  2. Please provide clear inclusion / exclusion criteria in methodology.
  3. Please provide primary and secondary outcomes of the study in methodology.
  4. Which scales were used for assessment of the pain? There is no even a word in methodology.
  5. There is nothing about statistical analysis in methodology. Statistical methods should be described in details.
  6. Baseline characteristics of the patients are too basic, more demographic data such as BMI, injured organs, mechanism of injury, comorbidities, ASA classification… should be presented.
  7. The authors found that pain score was assessed only in 20 out of 2432. Any further analyses with such weak sample size are meaningless. They compared RR, SBP, HR among the patients who had (n=20) and those who did not have (N=2412) pain score? This sample is not representative and is possible source of significant bias.
  8. Each abbreviation used in Table should be explained in the legend of the table.
  9. The presentation of the results is very wordy, not reader friendly with many unnecessary details. Also, there is significant amount of unnecessary analyses (E.g. Tables 6-9), which does not fit with the main objective of the study.
  10. Discussion is very extensive with many repetitions from existing literature.
  11. References should be revised (eg. ref No 2 – Author 1, A.; Author 2, B. Title of the chapter. In Book Title, 2nd ed.; Editor 1, A., Editor 2, B., Eds.; Publisher: Publisher Location…)

Reviewer 2 Report

The manuscript by Holak, Czapla, and Zielinska addresses an important issue, pain assessment and management in prehospital emergency medical services for children.  This is a critical area in which to highlight where improvements could be made.  The location of the study is very specific (Warsaw and surrounding area), but the finding have the potential to inform audiences everywhere.  The authors should be commended for their work on this study.  I have a few concerns/areas in which the manuscript could be improved.

Major Concerns:

A lot of information is missing from the Materials and Methods section, in particular related to the presentation and analysis of results. There are numerous tables (more on that next), yet there is nothing describing what breakdowns were of interest.  P-values are reported in several places (and identified there as, for example, chi-square test), but this should all be described in the Methods section.  There are a couple of instances where methods are introduced or repeated in the Results section.  These belong only in the Methods section.

There are numerous tables (16 in main paper).  Many of them have more than 5 rows and 5 columns.  This is too much tabular information.  Can the authors focus on the items of greatest interest/impact.  If needed, some of the tables could be moved to the Supplement (although there are already 6 more tables there).  It has the feeling of just cross-tabulating everything that was collected without a goal.

The Discussion section has quite a bit of literature review, but most of it lacks the tie-in with the current study (e.g., compare/contrast, differences between study designs/populations).

There are a number of grammatical errors, unclear phrases, and poor wording choices (e.g., "underappreciated" in Abstract for non-pharmacological methods; last sentence in Introduction seems to refer to the frequency of acute pain, but I think it's referring to the assessment and treatment methods).  The manuscript requires quite a bit of cleaning regarding the use of English.

Minor Concerns:

In the abstract (and a similar sentence in Results), it says that 1% of injured children had pain intensity rated and NRS was used in all patients.  I believe this to mean "all" of the 1% of patients, but it is difficult to decipher. 

Introduction - The description of S vs. B-EMTs is a bit hard to follow.  I am not really sure what one of these teams looks like in terms of medical training.

Table 1 - Overall summaries of accident site and injury type could also be included here.  Also, minimum age in months doesn't seem to match minimum age in years - 1.0 months corresponds to between 0.079 and 0.087 years (depending on rounding), which would always round up to 0.1 years.

Results, second paragraph - The first two age categories seem to overlap (0-1 and 1-3).  Injury type (low/high-energy) were never defined.

Results, Section 3.3 (line 147) - This states that "[IV] and rectal route [sic] were preferred".  I do not see how this follows from the information in the table.